# The Modulatory Action of Vitamin D on the Renin–Angiotensin System and the Determination of Hepatic Insulin Resistance

**DOI:** 10.3390/molecules24132479

**Published:** 2019-07-05

**Authors:** Po Sing Leung

**Affiliations:** School of Biomedical Sciences, Faculty of Medicine, The Chinese University of Hong Kong, Hong Kong, China; psleung@cuhk.edu.hk; Tel.: +852-3943-6879

**Keywords:** AT1 receptor calcitriol, HepG2 cells, liver, obesity, pancreas, type 2 diabetes

## Abstract

Vitamin D deficiency or hypovitaminosis D is associated with increased risks of insulin resistance, type 2 diabetes mellitus (T2DM) and its related non-alcoholic fatty liver disease (NAFLD). Meanwhile, inappropriate over-activation of the renin–angiotensin system (RAS) in the liver leads to the hepatic dysfunction and increased risk of T2DM, such as abnormalities in lipid and glucose metabolism. Our previous findings have shown that calcitriol, an active metabolite of vitamin D, reduces hepatic triglyceride accumulation and glucose output in diabetic db/db mice and human hepatocellular cell HepG2 cells under insulin-resistant conditions. Notwithstanding the existence of this evidence, the protective action of vitamin D in the modulation of overexpressed RAS-induced metabolic abnormalities in the liver under insulin resistance remains to be elusive and investigated. Herein, we have reported the potential interaction between vitamin D and RAS; and its beneficial effects on the expression and function of the RAS components in HepG2 cells and primary hepatocytes under insulin-resistance states. Our study findings suggest that hormonal vitamin D (calcitriol) has modulatory action on the inappropriate upregulation of the hepatic RAS under insulin-resistant conditions. If confirmed, vitamin D supplementation might provide a nutraceutical potential as a cost-effective approach for the management of hepatic metabolic dysfunction as observed in T2DM and related NAFLD.

## 1. Main Text

Hypovitaminosis D is commonly observed in many countries; it is known to be closely associated with increased insulin resistance, impaired insulin secretion, and poorly controlled glucose homeostasis, and thus is correlated with the risk of metabolic diseases, such as type 2 diabetes mellitus (T2DM) and non-alcoholic fatty liver disease (NAFLD). Meanwhile, the liver plays key roles in glucose and lipid metabolism, and its deregulation leads to abnormalities in hepatic glucose and lipid metabolism. In this context, the recent identification of vitamin D receptors (VDR) and of the vitamin-D activating 1α-OHase in several non-renal tissues, including the pancreatic islets and liver, explains the diverse physiological roles of vitamin D, in addition to ‘classic’ effects on calcium and bone homeostasis [1]. The presence of VDR response elements (VDREs) in promoter regions of insulin receptor genes may explain these non-classical effects of vitamin D [2]. Eighty percent of circulating insulin binds to insulin receptors in the liver, which occurs throughout the liver lobules [3]. Hepatic insulin resistance is a consequence of hepatic adiposity and accounts for the majority of the overall insulin resistance, independent of obesity [4]. Reduced hepatic insulin sensitivity in metabolic syndromes, such as type 2 diabetes (T2DM) and non-alcoholic fatty liver disease (NAFLD) [5], may result directly from reduced insulin receptor numbers or indirectly from reduced insulin action via post-insulin receptor signalling pathway genes [6]. It may also result from a reduction in Gc globulin (vitamin D binding proteins synthesized by hepatocytes and stored in stellate cells) through reduced megalin or gp330 receptor function [7].

It has been well recognized that hypovitaminosis D is associated with increased risks of NAFLD and hepatic fibrosis with cirrhosis developing in 10% of such patients [8]. In fact, hypovitaminosis D is associated with increased risks of each component of metabolic syndrome: central obesity, glycemia, blood pressure, adverse lipid profiles, insulin resistance, T2DM, and cardiovascular disease [9,10]. The effects of vitamin D on insulin sensitivity have yet to be investigated. In the pancreatic islets, it may be due, in part, to increased intracellular ionised calcium or modulation of the many genes with VDREs in their promoter regions [2]. NAFLD increases hepatic insulin resistance, independent of general obesity in humans [5] while vitamin D inhibits adipogenesis via VDR mediated inhibition of PPARγ activity [11]. Vitamin D is both 25-hydroxylated and activated in the liver by local 1α-hydroxylase and there are fully functional VDRs in Kupffer, stellate and endothelial cells as well as in the hepatocytes [12]. In view of this fact, vitamin D could be expected to exert direct effects on hepatic insulin signalling pathway genes and vitamin D supplementation might have a potential for the management of obesity-associated T2DM and NAFLD [1].

The renin–angiotensin system (RAS) is best known as a hormonal system with a physiological role in the maintenance of circulatory and fluid homeostasis, i.e., a classical hormonal system associated with human hypertension and diabetes. Dysregulation of the RAS results in cardiovascular and renal disease as well as metabolic syndromes [13]. Sequential enzymatic cleavage of the circulating renin and angiotensin-converting enzyme (ACE) on the liver angiotensinogen produces angiotensin II, the physiologically active peptide of the system, acting mainly on its respective AT1 and AT2 receptors. In addition, alternative enzymes (e.g., ACE2 and chymase) are able to produce a number of bioactive peptides, such as angiotensin IV, Ang (1–7), and Ang (1–9) [13]. Apart from the classical RAS, local RASs have been identified in various cells and tissues, including the pancreas and liver [14].

Interestingly, it has been previously reported that hypovitaminosis D is associated with increased blood pressure while vitamin D reduces RAS activity through specific suppression of renin secretion [15]. Paradoxically, a fall in systolic blood pressure with vitamin D supplementation in T2DM was associated with falls in angiotensin II but increases in renin in the circulation, suggestive of an alternative mechanism involved [16], probably via a reduction in vascular resistance by vitamin D directly [17]. It has also been further reported that vitamin D is a negative regulator of the RAS and relevant to the regulation of hypertension and cardiovascular disease through downregulation of renin secretion; in addition, RAS activity is increased markedly in VDR-null mice [18,19].

More interestingly, we have recently identified a functionally active islet RAS in the pancreas [20] and the expression and function of this local RAS are also subject to the modulatory action of vitamin D [21]. Indeed, calcitriol, an active form of hormonal vitamin D, can prevent and correct overactive RAS in isolated islets under high-glucose conditions; these effects are in parallel with the well-known action of calcitriol on increasing islet beta-cell glucose-stimulated insulin secretion [20]. Furthermore, we have also demonstrated that mice with diet-induced hypovitaminosis D develop impaired glucose tolerance, increased RAS expression and decreased islet function-related gene transcription [22]. Pharmacological treatment with aliskiren, a renin inhibitor, without vitamin D status correction, has been found to reduce islet RAS overactivity, islet dysfunction and insulin resistance while improving glucose tolerance [22]. These study findings point to RAS inhibition and/or vitamin D supplementation being a potential for ameliorating islet dysfunction and insulin resistance observed in T2DM and NAFLD. In term of the liver, the existence of key components of the RAS while their upregulation and clinical relevance to hepatic dysfunction, such as liver fibrosis, have long been recognized [23,24]. As expected, there is clear evidence for the association of hypovitaminosis D with hypertension and RAS over-activity, whilst vitamin D treatment can lower human blood pressure by suppressing renin formation, reducing RAS activity. On the other hand, RAS blockade appears to be protective against T2DM and NAFLD whilst vitamin D supplementation reduces insulin resistance and dysfunction in the liver and pancreas [1].

Interestingly, our more recent study has demonstrated that vitamin D can protect hepatic function in the context of glucose and lipid metabolism via the mediation of Ca2+/CaMKKβ/AMPK signaling [25]. Despite these findings, the potential interaction between the RAS and vitamin D in the modulation of hepatic glucose and lipid metabolism has yet to be determined. As expected, our preliminary results showed that major RAS components (e.g., angiotensinogen, AT1/AT2/Mas receptors, and ACE/ACE2) were expressed in the HepG2 cells and isolated hepatocyte cells. Our preliminary data also further showed that high-glucose plus high-insulin culture conditions significantly upregulated the expression of the key RAS components, notably the AT1 receptor in HepG2 cells (unpublished data). In addition, treatment with different concentrations of calcitriol (0.01–10 nM) significantly suppressed the inappropriate activation of the AT1 receptor expression under insulin-resistant state in a dose dependent manner; these inhibitory effects on AT1 receptor were observed from 4 to 6 hours after calcitriol treatment (unpublished data). Interestingly, the beneficial effects of vitamin D on the RAS may be mediated via AMPK-Sirt1 signaling pathway, as demonstrated by the pharmacological use of SIRT1 inhibitor, EX-527 (unpublished data). In corroboration with the HepG2 cells, similar results on the protective effects of calcitriol were also observed in the isolated hepatocytes. These data prompt us to speculate that calcitriol (an active form of vitamin D) has modulatory action on the inappropriate activation of the hepatic RAS under insulin-resistant conditions, and that potential vitamin D supplementation might provide a cost-effective measure for improving hepatic metabolic dysfunction in T2DM and related NAFLD.

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
