# Peer review of "The Modulatory Action of Vitamin D on the Renin–Angiotensin System and the Determination of Hepatic Insulin Resistance"

_molecules, 2019, doi:10.3390/molecules24132479_

Round 1

Reviewer 1 Report

The commentary of Leung Po Sing that put light on a new biochemical pathway involved in the intricate T2DM and related NAFLD mechanisms. Double space at line 90 after the dot.

Author Response

Double space at line 90 after the dot has now been given.

Reviewer 2 Report

Major comments

This is an interesting commentary ,based on available pathophysiological evidence on animal models.Studies included are presented in the context of the potential of vitamin D supplementation for the management of hepatic insulin resistance in T2DM and NAFLD.

From my perspective, there is an initial pathophysiological backrground,associating hypovitaminosis D ,with insulin resistance states.THe main focus of this commentary is on the 

beneficial effects of calcitriol on overactive RAS in rat pancreatic islets.However,this effect,has not been shown to improve insulin secretion and glucose homeostasis.Although,an indirect effect might be hypothesized,further data (mainly in human islets )would provide the missing link ,between the observed biophenomena.Until then,vitamin D supplementation for T2DM and NAFLD is not considered as a potential clinical approach,based on available RCTs (showing contradicting findings).I would suggest some revision of this commentary to this direction.

Specific comments

Main text:Please use hypovitaminosis D instead of ..its deficiency..Line 27.Please also provide references fro your statements,since available systematic reviews ,do not support supplementration to be effective ,with the exception of profound hypovitaminosis D .(30-31).

I would suggest starting the text from the physiological backrgound (i.e. VDR receptors in liver,pancreas,cicrulating insulin etc) then go to pathophysiology and then report the current state of clinical evidence.This would provide ,a more realistic approach,since vitamin D supplementation has been largely ineffective for CVD,T2DM so far.I would also avoid phrases such as ..(lines 49, 50), since the picture is not clear so far.Associations are there but ,there is minimal or no effect in RCTs.May be the author would like to report some of the reasons for this gap.The author also reports interesting results from his group,however,no recent findings on RAS activity reduction by vitamin D supplementation.The most recent study is on reduction of hepatic accumulation and not RAS activity.Expected unpublished results will be very interesting on this topic.

Minor points

Typos:line 70 previsously

Refs 11 and 12 are not separated

Line  63:Is there any other more appropriate ref for this statement?

Line 70-72 .Has this been confirmed by other studies?

Author Response

The comments have been adequately taken and included with modifications in the revised version, as suggested.

Reviewer 3 Report

Summary:

The present paper reflects the author' efforts to support their hypothesis of that the activated form of vitamin D (calcitriol) could modulate the activity of hepatic RAS and the hepatic glucose metabolism, therefore vitamin D supplementation may become a possible alternative prophylactic and/or therapeutic agent in the prevention or treatment of T2DM or NAFLD.

The study is interesting and meaningful, and the author contribute to this field of research with new findings, nevertheless some minor revisions are required before publication.

1. While the paper is original and thorough, I think that an extensive revision and reformulation is required for the whole manuscript. The author need to rephrase several paragraphs of the manuscript.

2. Regarding to argumentation in the sentence starting at line 108. “Interestingly, the beneficial effects of vitamin on the RAS may be mediated via AMPK-Sirt1 signaling pathway.”, the author should supplement to support the reasoning basis.

3. Line 79 and 80: “More interstingly, we have recently identified a functionally active islet RAS in the pancreas [19], of which its expression and function are subject to the modulatory action of vitamin D [20]”. To be more clear and accurate please, rephrase the sentence.

4. Nevertheless, the references are adequate, and up to date, the number 12 Reference appear twice in the reference list. However both of the references are revealed for the paragraph, please, verify and correct the mistake.

Author Response

1. The potential involvement of SIRT1 has now been supplemented so as to support the reasoning basis.

2. The sentences has been rephrased, as suggested.

3. It was done.

Round 2

Reviewer 3 Report

Regarding to the AMPK-SIRT1 pathway, the author added a short explanation, without a relevant reference to support the affirmation. Please, insert a reference to this sentence.